# Application of Ultra Narrow Band Modulation in Enhanced Loran System

**DOI:** 10.3390/s21134347

**Published:** 2021-06-25

**Authors:** Boyun Lyu, Yu Hua, Jiangbin Yuan, Shifeng Li

**Affiliations:** 1National Time Service Center, Chinese Academy of Sciences, Xi’an 710600, China; hy@ntsc.ac.cn (Y.H.); yuanjiangbin@ntsc.ac.cn (J.Y.); lishifeng@ntsc.ac.cn (S.L.); 2University of Chinese Academy of Sciences, Beijing 100049, China; 3Key Laboratory of Precise Positioning and Timing Technology, Chinese Academy of Sciences, Xi’an 710600, China

**Keywords:** data rate, enhanced Loran (eLoran) system, extended binary phase shift keying (EBPSK) modulation, ultra narrow band (UNB) modulation

## Abstract

The Enhanced Loran (eLoran) system is valued for its important role in the positioning, navigation, and timing fields; however, with its current modulation methods, low data rate restricts its development. Ultra narrow band (UNB) modulation is a modulation method with extremely high spectrum utilization. If UNB modulation can be applied to the eLoran system, it will be very helpful. The extended binary phase shift keying modulation in UNB modulation is selected for a detailed study, parameters and application model are designed according to its unique characteristics of signal time and frequency domains, and it is verified through simulation that the application of this modulation not only meets the design constraints of the eLoran system but also does not affect the reception of the respective signals of both parties. Several feasible schemes are compared, analyzed, and selected. Studies have revealed that application of UNB modulation in the eLoran system is feasible, and it will increase the data rate of the system by dozens of times.

## 1. Introduction

The Enhanced Loran (eLoran) system is the best land-based backup for space-based satellite navigation systems. This is a key technical means for space-based and ground-based integrated navigation applications. It is also an important information source for the construction of a comprehensive national positioning, navigation, and timing (PNT) system. It plays an important role in national economy and defense security. In recent years, the eLoran system has continuously garnered increasing attention [1,2,3].

The eLoran system adds a data channel based on the traditional Loran-C pulse signal and conveys basic time, warnings, corrections, and integrity information to the users through modulation methods [4]. The existence of this data channel allows the eLoran system to use non-precision instruments to meet the demanding requirements of aircraft landing, and to safely navigate the ship to the port under low visibility conditions. The data channel can also provide differential data for the backup of global satellite navigation system, or provide modified data such as the additional secondary factor to further improves the timing accuracy of the eLoran system [5,6,7,8]. The current main modulation methods include the Eurofix [9] and ninth pulse modulation [10]; the achievable data rate is 7 bits for every group repetition interval (GRI), approximately 70–175 bps and 5 bits for every GRI, approximately 50–125 bps, the data rate is lower when more feature information need to be transmitted, and in some applications, such as the eLoran system as the land-based supplementary channel of the wide area augmentation system; the minimum data rate required for transmission is 250 bps [11]; and as a conceptual prototype of the low frequency/medium frequency alternative PNT (LF/MF APNT) system proposed by the United States Federal Aviation Administration APNT Working Group in August 2010, the minimum data rate required for the system is 1500 bps [12], the current data rate is far from insufficient. Therefore, the domestic and foreign researchers are actively developing new solutions [13,14,15]. Improving the data communication capabilities of the eLoran system can be regarded as the technical focus of the next-generation eLoran sytem, this technological breakthrough can not only strengthen the existing system functions, but also meet more application requirements in the future, which has important theoretical and practical value.

Ultra narrow band (UNB) modulation is a modulation method with an extremely high spectrum efficiency. It was firstly conceived by the American engineer H. R. Walker and then the modulation idea has gradually derived a series of high-efficiency modulation techniques, like the very minimum shift keying, pulse position phase reversal keying, extended binary phase shift keying (EBPSK) modulation and so on [16,17,18,19,20]. The modulation can perform high-speed data transmission within a very narrow signal bandwidth, which can reach more than 30 bps/Hz [21]. It has been continuously applied in some new fields in recent years and the receiving methods are highly innovative and developed [22,23,24,25,26]. If this technology can be applied to the eLoran system, the shortcomings of the eLoran system will be compensated. Therefore, based on the idea of the fusion application, this paper carries out the following specific research work.

The remainder of this paper is organized as follows. Section 2 presents the signal characteristics of the eLoran system, the design constraints, the basic principles of the UNB modulation and its parameter design. Section 3 explores the feasibility of the fusion between the eLoran system and the UNB modulation, also gives some schemes to compare. Section 4 discusses the results and the future research directions. Section 5 summarizes the full text and concludes that the application of UNB modulation in the eLoran system is feasible, and it can be used as a solution to improve the data rate of the eLoran system.

## 2. Materials and Methods

eLoran has a unique pulse signal system and waveform characteristics, when applying some new signal systems in its frequency band, certain design constraints must be met. UNB modulation is a special modulation mode, which also has its unique signal characteristics. In the realization of fusion application with eLoran system, detailed design of modulation parameters is required. These contents will be introduced in turn in this section.

### 2.1. eLoran

eLoran uses a standard Loran-C pulse signal system. The pulse waveform has a strict physical definition. It contains more than 99% of the signal energy in the 90–110 kHz frequency band, and the center frequency is 100 kHz. This is a low-frequency radio transmission system. The time-domain signal waveform and spectral characteristics are as shown in Figure 1.

The system adopts the pulse group signal transmission method, the main station is a group of nine, the substation is a group of eight, the first eight pulses are separated by 1 ms, the eighth and ninth pulses are separated by 2 ms, and the pulse GRI ranging 40,000–99,990 μs, is selected with steps of 10 μs, with a different selection between each chain. Taking the BPL (The call sign of eLoran time service system in China) broadcaster station as an example, it uses the form of a master station to transmit signals, and has a GRI of 60 ms, as shown in the Figure 2.

### 2.2. Design Constraits

To adopt a new signal system in the Loran-C signal frequency band, relevant regulations must be observed, which mainly include Loran-C signal protection and harmonic suppression requirements.
Protection criteria

The standard ITU-R M.589-3 “Technical characteristics of methods of data transmission and interference protection for radionavigation services in the frequency bands between 70 and 130 kHz” indicates that there are various services, including radionavigation systems, operating in frequency bands from 70 to 130 kHz; the radionavigation is a safety service, and all practical means consistent with the radio regulations should be adopted to prevent harmful interference to any radionavigation system [27].

The protection criteria for the Loran-C/CW interference (CWI) as a function of frequency offset are as shown in Figure 3. From Figure 3, the maximum unwanted-to-wanted signal level is −20 dB. 

The Chinese Standard GB 13613-2011 “Electromagnetic environment requirement for sea long range radio navigation stations and monitors” also has the similar protection criteria rules [28].
Harmonic suppression criteria

In Chinese Standard GB/T 14379-93 “Generic specification for Loran-C system”, regulations are also made for harmonic suppression [29].

Relative to the fundamental wave of 100 kHz, the spectral line average of each harmonic does not exceed the regulations listed in Table 1.

### 2.3. UNB Modulation

Considering the extreme case of a narrowband, the frequency spectrum of the cosine signal is a non-zero spectral line located at the carrier frequency. Theoretically, the bandwidth is zero, and the energy is highly concentrated; however, the signal cannot transmit any useful information. The basic principle of the UNB modulation is to slightly dither the amplitude, phase, shape, or symmetry of the cosine signal, such that the spectrum energy is still highly concentrated at the carrier frequency, and the bandwidth is ultra-narrow. Using a special “zero-group delay filter”, the carrier frequency can pass smoothly, out-of-band noise is greatly suppressed, signal modulation feature is prominent, and demodulation is realized.

The EBPSK modulation in the UNB modulation was selected for specific research, and the time domain expression was as follows [30]:(1)f0(t)=Asin2πfct,       0≤t<Tf1(t)={Bsin(2πf ct+θ),    0≤t<τAsin2πfct,    τ≤t<T,
where, $f_{0}\left(t\right)$ and $f_{1}\left(t\right)$ are the modulation waveforms in [0, T] when sending symbol “0” and symbol “1”, $f_{c}$ is the carrier frequency, B and A are the signal amplitudes during phase jump or not, θ is the phase jump angle, also known as the modulation angle, 0≤θ≤π, T and τ are respectively the symbol period and modulation interval, $T=N/f_{c}$, lasts for N≥1 carrier period, $\tau =K/f_{c}$, lasts for K<N carrier period, N and K are integers.

In a previous study, the expression of the power spectral density of a general binary equal-probability EBPSK modulated signal was derived based on the Wiener–Khinchin theorem as follows [31]:

When $f=f_{c}$, the component at the carrier frequency is as follows:(2)$$S_{EBPSK}\left(f_{c}\right)=\frac{\left(A^{2}+B^{2}-2ABcos\theta \right)\tau^{2}}{16T}+\frac{\left[A^{2}{\left(2T-\tau \right)}^{2}+B^{2}\tau^{2}+2AB\left(2T-\tau \right)\tau cos\theta \right]}{16T^{2}}\delta \left(0\right)$$

When $f\neq f_{c}$, the continuous spectrum in the first half and harmonic components in the second half are as follows:(3)$$\begin{array}{cc} S_{EBPSK}\left(f\right)= & \frac{\left[{\left(Bcos\theta -A\right)}^{2}f_{c}{}^{2}+B^{2}f^{2}sin^{2}\theta \right]\left(1-cos2\pi f\tau \right)}{8T\pi^{2}{\left(f_{c}{}^{2}-f^{2}\right)}^{2}}\\ & \;+\sum_{\overset{m=-\infty }{\underset{m\neq \pm N}{\;}}}^{\infty }\frac{\left[{\left(Bcos\theta -A\right)}^{2}f_{c}{}^{2}+B^{2}{\left(\frac{m}{T}\right)}^{2}sin^{2}\theta \right]\left(1-cos2\pi \frac{m}{T}\tau \right)}{8T^{2}\pi^{2}{\left(f_{c}{}^{2}-{\left(\frac{m}{T}\right)}^{2}\right)}^{2}}\delta \left(f-\frac{m}{T}\right) \end{array}$$

It is observed that the continuous spectrum component is affected by the factor (1−cos2πfτ). The power spectrum produces zero points with a period of $1/\tau =f_{c}/K$. The main lobe width of the power spectrum is $2f_{c}/K$, whereas the side lobe width is $f_{c}/K$, and decreases as the value of K increases. The discrete spectrum appears at intervals of frequency $1/T=f_{c}/N$, and the interval frequency decreases as the value of N increases. In addition, the discrete spectrum is affected by the factor $\left(1-cos2\pi \frac{m}{T}\tau \right)$ function and disappears when the continuous spectrum becomes zero.

Let A=B, θ=π, N=20, K=1, carrier frequency $f_{c}=50\;\mathrm{kHz}$, and sampling frequency $f_{s}=2\;\mathrm{MHz}$, we randomly generate 30,000 data symbols with equal probability. The time-domain waveform and spectral characteristics of the EBPSK modulated signal are as shown in Figure 4. It is observed that the energy is highly concentrated at the carrier frequency, and the bandwidth is ultra-narrow. The main lobe width is 100 kHz, the side lobe width is 50 kHz, and the power spectral density is nearly zero at 100 kHz.

### 2.4. Parameter Design

According to the spectral characteristics of the EBPSK modulated signal, it was assumed that if the signal’s PSD at 100 kHz and its harmonics were nearly zero, the modulation may meet the design constraints of the eLoran system, and thereby realizing the fusion application; thus, the parameter design was needed.

The EBPSK modulation design parameters mainly included the number of symbol cycles N, number of modulation cycles K, modulation angle θ, signal carrier frequency $f_{c}$ and the signal amplitude A and B. The following is an analysis and description of the choices in these aspects.

#### 2.4.1. General Parameters

The general parameters refer to N, K, and θ, and the literature [32,33,34] analyzes their values in detail.
N: When other parameters remain unchanged, the larger the value of N, the more concentrated the signal energy, and the narrower the EBPSK modulated signal bandwidth; however, for a large value of N, the number of symbols transmitted per unit time will decrease, resulting in a decrease in the data rate, and the data rate calculation formula is Equation (4). In addition, considering that the signal has a specific recovery time after passing through the zero-group delay filter, it is usually designed to make N≥10 [35].
(4)$$Data\;Rate=1/\left(\frac{N}{f_{c}}\right)=\frac{f_{c}}{N},$$K: On the premise that other parameters remain unchanged, the value of K increases, difference between the two symbol waveforms becomes larger, width of the main and side lobes of the EBPSK modulated signal power spectrum becomes narrower, amplitudes of the main lobe and adjacent side lobes increase slightly, and energy becomes more dispersed. To ensure a narrower bandwidth in the design, the K value is generally 1–2, which is small.θ: When other parameters are fixed, the smaller the value of θ, the more concentrated the power spectrum energy of the EBPSK modulated signal, and the smaller the occupied bandwidth. Studies have revealed that when θ=π, the sideband is approximately −24 dB lower than the carrier frequency, and if the power level of the sideband is lower than the carrier frequency by approximately −40 dB, θ≤π/6 is required, but the signal demodulation performance will be reduced accordingly. To ensure the demodulation performance, θ=π is often used.

#### 2.4.2. Application Parameters

The application parameters include $f_{c}$ and A and B, which will be selected and designed according to different applications. In the application of the eLoran system, the specific design process was as follows:$f_{c}$: First, it was clear that the value should be selected in the long-wave band 30–300 kHz, such that the EBPSK modulated signal could be shared with the eLoran signal for the same transmitter and antenna, and the field strength calculation formula is also the same [36]; second, the power spectral density of the EBPSK modulated signal at 100 kHz frequency was required to be 0, and according to the introduction of the characteristics of Section 2.3, it was observed that the required Equation (5) was obtained; third, attempt to prevent it from attaining 100 kHz or its harmonic frequency; only in this way could the eLoran system design constraints be better met, and the fusion application can be realized.
(5)$$\frac{2f_{c}}{K}+\frac{nf_{c}}{K}=100\;\mathrm{kHz},\;n\geq 0,$$A
and B: in the design of EBPSK modulation parameters, let A=B and θ=π to form the antiphase modulation, and let A=0 (also known as missing cycle modulation) or B=0, which is known as narrow pulse modulation, similar to radar. Here, we selected the uninterrupted signal A=B situation, and set the ratio of their amplitude to the peak amplitude of the Loran-C signal to η; thereafter, according to the difference in the power spectral density between the EBPSK modulated and Loran-C signals (in dB) meeting the design limitation conditions, the η value was adjusted continuously until it was satisfied. The value of η was the maximum ratio of the amplitude of the two signals, denoted as η_max_, that is, the maximum value of A and B is 1/η of the peak amplitude of the Loran-C signal.

The parameters of Figure 4 and the other figures in the rest of the paper about EBPSK modulation were both designed as the regulations mentioned above.

## 3. Results

This section is the focus of the full text. The simplified analysis model of the adopted UNB modulation method applied to the eLoran system is as shown in Figure 5.

In Figure 5, the transmission system can be divided into two branches: the Loran-C signal and UNB modulation branches. The two branches are directly superimposed and transmitted. The channel adopted the Gaussian white noise channel, and at the receiving end, the Loran-C signal branch passed through the bandpass filter to obtain the signal. The UNB modulation branch mainly used an impacting filter for demodulation to obtain random sequence information.

Whether the system was feasible or not, it was first necessary to analyze whether the newly added UNB modulation branch met the design constraints in Section 2.2; second, we analyzed the degree of mutual influence between the two branch signals, which could be determined according to the signal reception situation at the receiving end. The specific analysis is provided in the following subsections.

### 3.1. Satisfaction of Design Constraints

The parameters used in Figure 4 are as follows: A=B, $f_{c}=50\;\mathrm{kHz}$, θ=π, N=20, K=1, and $f_{s}=2\;\mathrm{MHz}$. Let the EBPSK modulated signal and Loran-C signal peak amplitude ratio be η=1/20, the simulation time was 30 GRI = 30 × 60 ms = 1.8 s, and the signal power spectrum generated by the pwelch function was as shown in Figure 6, where the Hamming window used was 2000, which is the length of a Loran-C pulse.

Figure 6a shows the respective power spectral densities of the two branch signals, we can see that they were significantly different around 100 kHz, which seemed to meet the protection criteria of the eLoran system. Figure 6b shows the difference or ratio between them, in dB, from which the specific unwanted-to-wanted signal ratio at each frequency point could be analyzed more clearly to check whether they meet the criteria [36]. Because the ratio value in the 30 kHz range on the left of 100 kHz was slightly larger than that on the right, it was only necessary to investigate whether the ratio of each frequency point in the range 70–100 kHz meets the requirements. The key frequency points shown in the Figure 6b were compared with those of Figure 3, and it was observed that the ratio of each frequency point in the range 70–100 kHz met all the protection requirements.

Reconsidering its harmonic suppression level, as shown in Figure 6a, the spectral line level of the EBPSK modulated signal at 200 kHz was approximately 85 dB lower than that of the 100 kHz fundamental wave level of the Loran-C signal, and it was approximately 95 dB lower at 300 kHz, higher harmonics lower; thus, the system design also met the proposed harmonic suppression criteria.

### 3.2. Influence of Signal Reception on Each Other

After meeting the design constraints, the next step was to analyze the degree of mutual influence between the two branches in the specific communication process; particularly, the EBPSK modulation could not affect the existing Loran-C signal.

Continuing the simulation of the receiving end, the Loran-C branch used a 511th-order bandpass finite impulse response filter with a lower cutoff frequency of 85 kHz and a higher cutoff frequency of 115 kHz, with a sample rate of 2 MHz. The signal waveform before and after filtering and the comparison with or without the ultra-narrowband modulation signal after filtering are as shown in Figure 7.

As shown in Figure 7a, the blue and red lines represent the signal waveforms before and after filtering, respectively. By observing the pulse signal, it can be obtained that the bandpass filter caused a specific time delay. By observing the period of pulse-free signal, it can be concluded that the bandpass filter could basically filter out the EBPSK modulated signal without affecting the reception of Loran-C signal. Furthermore, the Loran-C signal without superimposed EBPSK modulated signal passing through the same bandpass filter resulted in the same time delay. Comparing the filtered pulse signals of the two cases for the same time period, as shown in Figure 7b, it was observed that the signal waveforms were consistent, indicating that the existence of the EBPSK modulated signal did not affect the reception of the Loran-C signal.

The EBPSK branch at the receiving end was used to study the influence of the Loran-C signal on EBPSK modulated signal reception. Because the impacting filter used was designed based on 10 times the carrier frequency, the received signal first had to be down-sampled to 10×50 kHz=500 kHz; thereafter, the subsequent signal processing process was performed.

The design parameter values of the impacting filter were as Equation (6) and the corresponding frequency response characteristic is shown in Figure 8. It had the characteristic of “zero-group delay filter” designed by Mr. H.R. Walker using quartz crystal, that is, the phase change of the filter at carrier frequency was almost zero. This is a digital realization. The filter could filter out noise while highlighting the modulation angle change of the input signal as the amplitude jump of the output signal; thereafter, the threshold level setting was used to determine the amplitude jump to realize demodulation, when the amplitude jump occurs, it was judged as code “1”, otherwise, it was judged as code “0” [26].
(6)$$H\left(z\right)=\frac{b_{0}+b_{1}z^{-1}+b_{2}z^{-2}}{a_{0}-a_{1}z^{-1}-a_{2}z^{-2}-a_{3}z^{-3}-a_{4}z^{-4}-a_{5}z^{-5}-a_{6}z^{-6}},$$
where,


b=[1,−1.6181733185991785,1];



a=[1,−4.578193199274645,9.654665924115726,−11.692079480819313,…



8.5756341567768217,−3.6121554794765309,0.70084076007371199].


According to the analysis model shown in Figure 5, the EBPSK branch first converted the received superposed signal through the impacting filter into an amplitude jump signal s_1_, then took the absolute value of the s_1_ to generate s_2_, and finally got the amplitude jump envelope signal s_3_ through a lowpass filter and the filter adopted the Kaiser window, passband frequency was set to 10 kHz, stopband frequency was 20 kHz, passband ripple was 0.5 dB, stopband attenuation was 65 dB, the processed signal waveform s_3_ as the demodulation signal for the next threshold decision was as shown in Figure 9.

Figure 9a shows the 400 ms data intercepted, the blue and red lines are the processed signal s_3_ and the unprocessed received signal, respectively. It is observed that the received signal had a stable period of approximately 10 ms after the impacting filter. In actual use, the period could be discarded, and the remainder was used for normal demodulation. Continuing to intercept shorter data to make the results clearer, as shown in Figure 9b, simultaneously the waveform with only the EBPSK modulated signal at the same period was added. Comparing the three waveforms, it was observed that the amplitude change of the envelope after the signal processing only reflected the phase change moment in the EBPSK modulation, and it had no relevance with the appearance of the Loran-C signal. Therefore, it was concluded that the existence of the Loran-C signal did not affect the reception of the EBPSK modulated signal.

In conclusion, the EBPSK ultra-narrow-band modulation not only met the design constraints of the eLoran system but also did not affect the reception of Loran-C signal, and its reception and demodulation were simultaneously not affected by Loran-C signal; thus, the EBPSK ultra-narrow-band modulation and the eLoran system could be effectively integrated for application.

### 3.3. Schemes for Comparison

On the premise that the integrating application was feasible and all the above parameter design requirements were met, the following schemes were listed for comparison:

When listing the schemes, according to theoretical calculations, regardless of whether the value of K was 1, 2, or higher, there existed $f_{c}=50\;\mathrm{kHz}$ to meet the carrier frequency design requirements; thus, this carrier frequency selection was more adaptable.

All data rates of the schemes were calculated when EBPSK modulated signal was transmitted as an independent channel, rather than placed after the Loran-C pulse group in a GRI. Because as shown in Figure 9a, there was a period of stability time after the EBPSK modulated signal passed through the impacting filter, and the data in this period could not be used. As time elapsed, this situation no longer occurred in the later period; however, if the EBPSK modulated signal was placed after each Loran-C pulse group, each GRI had such a stable period, resulting in a waste of data rate.

Some conclusions can be drawn from the Table 2.
Comparing schemes one and two or schemes two and three, it was found that when K/N was smaller, the energy was more concentrated, and η_max_ could be larger, which implies that the EBPSK modulated signal had stronger anti-noise/interference ability and higher reliability.Comparing the first and third schemes, when the value of K was different, the position of 100 kHz in the power spectrum of the EBPSK modulated signal was different, and the value of η_max_ was greater for the position at the higher side lobe width of the EBPSK modulated signal power spectrum than that at the main lobe width or lower side lobe width.A comparison of the third and fourth schemes demonstrated that when the general parameters were similar, and the further away from the 100 kHz carrier frequency scheme, the greater the value of η_max_ and the higher the reliability, but the data rate was affected and the effectiveness was reduced.

In conclusion, considering both reliability and effectiveness, it was preferable to select option 2, that is, the simulation verification parameters used in the above study.

## 4. Discussion

A detailed analysis of the respective signal characteristics of the UNB modulation and eLoran system confirms that the modulation method meets the design constraints required by the system, and the simulation of their respective signal reception conditions confirms that the two will not interfere with each other. This demonstrates that the UNB modulation can be applied to the eLoran system.

The feasibility of this application scheme provides a solution for the increase in the eLoran system data rate, which can increase the current data transmission volume several times compared with the traditional methods like Eurofix or ninth pulse modulation. Besides, they can be used together and will not affect each other, such that more improved or enhanced data can be sent, providing a guarantee for eLoran system function stability and performance improvement, and more possibilities for the expansion of eLoran system applications.

UNB modulation can also be combined with channel coding measures or the improvement of its receiving demodulation decision method to further improve reliability [37]. Therefore, it is possible to select a solution with a higher data rate to achieve a win-win of effectiveness and reliability. This is the next step in the key research work.

## 5. Conclusions

UNB modulation has the advantage of high spectrum utilization. Combined with its special signal characteristics, it is applied to the eLoran system on the bases of meeting the design constraints to improve the data rate. This scheme is different from the existing pulse modulation, that is, modulation on the signal frequency band of eLoran system and Loran-C pulses. Instead, it is a modulation that uses a LF continuous wave different from the carrier frequency of eLoran system for data transmission and it is a bran-new technical solution to increase the data rate of the system. The scheme has the advantages of being simple to implement and can improve the data rate without too much modification of the existing eLoran system. It also provides an idea for the technical upgrade of the next-generation eLoran system. After this paper, the scheme will be verified on the experimental platform, in order to verify the feasibility and effectiveness of the scheme and provide a reference for the improvement of the scheme.

## Figures and Tables

**Figure 1 sensors-21-04347-f001:**
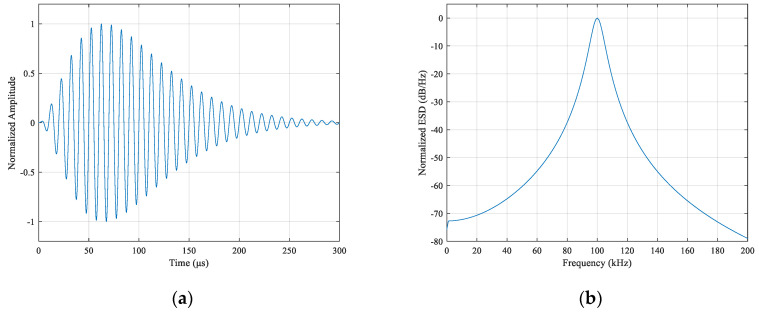
Characteristics of standard Loran-C pulse signal: (**a**) time-domain waveform; (**b**) energy spectral density (ESD) waveform.

**Figure 2 sensors-21-04347-f002:**
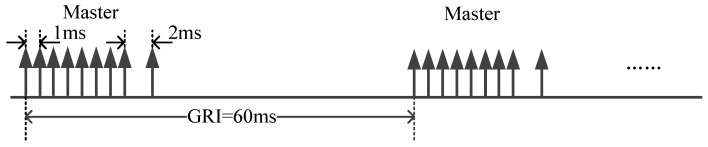
BPL station pulse group transmission form.

**Figure 3 sensors-21-04347-f003:**
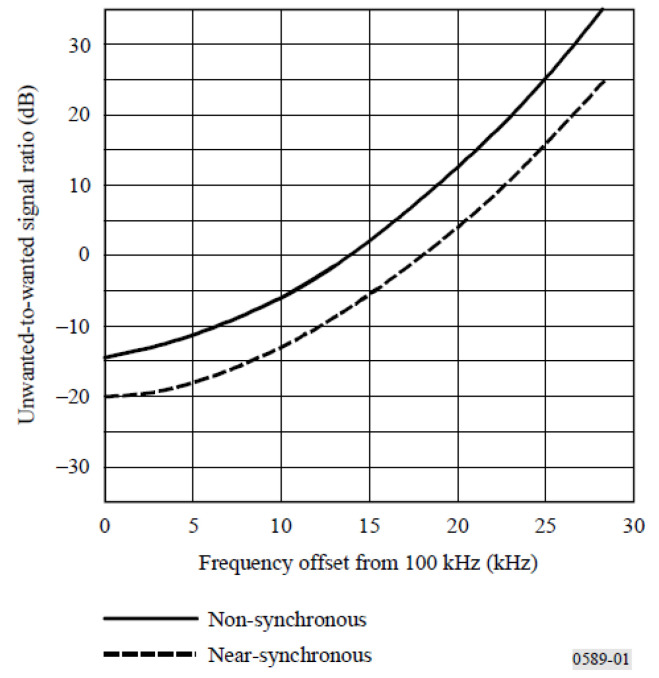
Loran-C/CWI protection criteria.

**Figure 4 sensors-21-04347-f004:**
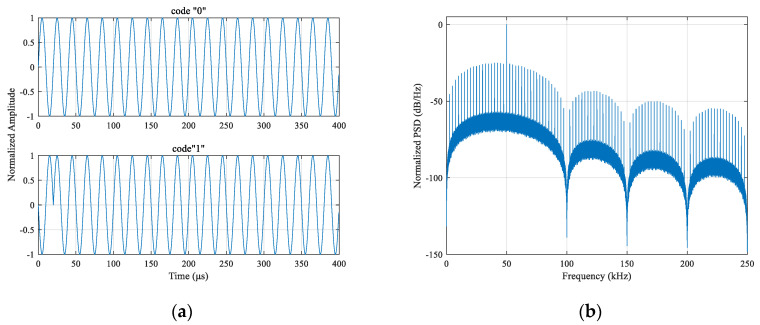
Characteristics of EBPSK modulated signal: (**a**) time-domain waveforms; (**b**) power spectral density (PSD) waveform.

**Figure 5 sensors-21-04347-f005:**
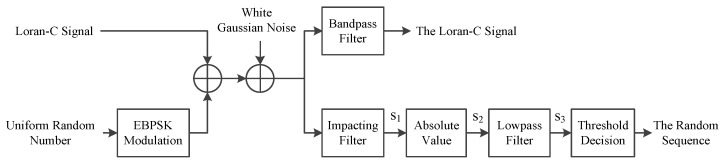
Simplified analysis model of using UNB modulation in eLoran.

**Figure 6 sensors-21-04347-f006:**
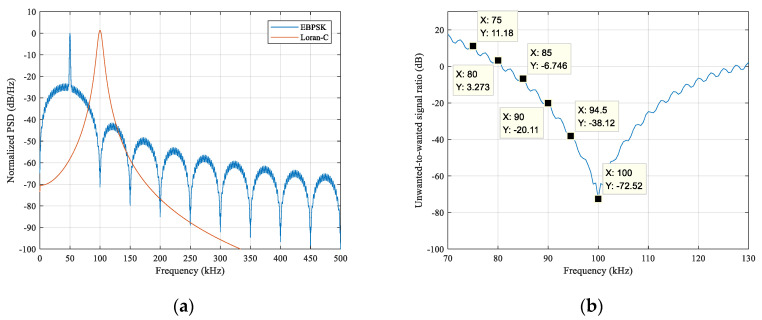
PSD of EBPSK modulated and Loran-C signals and their difference values: (**a**) PSD waveforms respectively; (**b**) difference between the EBPSK modulated and Loran-C signal.

**Figure 7 sensors-21-04347-f007:**
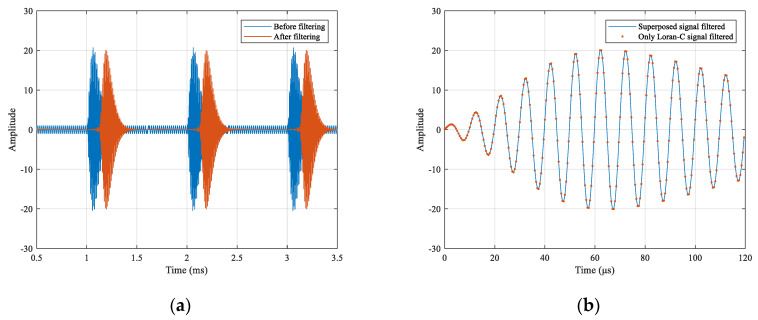
Comparison of Loran-C branch signals: (**a**) Comparison superposed signal before and after filtering; (**b**) Comparison with or without EBPSK modulated signal after filtering.

**Figure 8 sensors-21-04347-f008:**
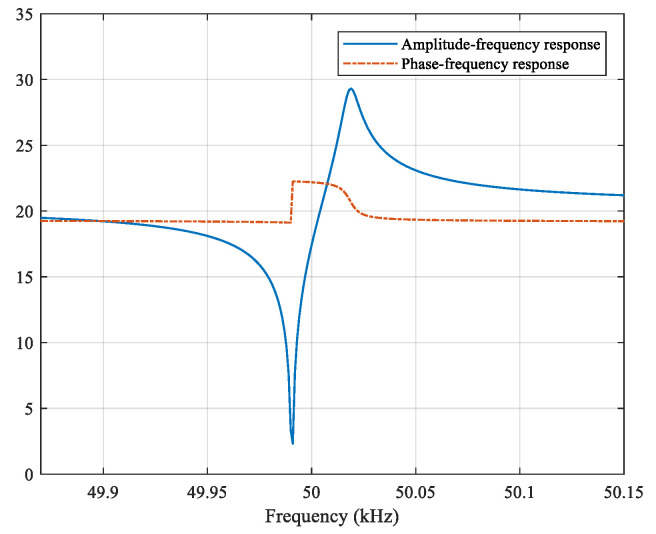
Frequency response characteristic of the impacting filter.

**Figure 9 sensors-21-04347-f009:**
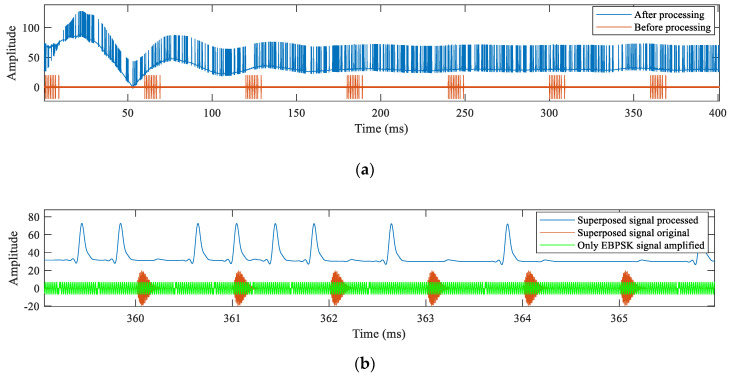
Comparison before and after processing of EBPSK branch signal: (**a**) comparison superposed signal before and after processing; (**b**) comparison after zooming in.

**Table 1 sensors-21-04347-t001:** Upper limit of each harmonic relative to the fundamental wave.

Harmonic Wave	Upper Limit Value, dB
2nd harmonic	−64
3rd harmonic	−68
4th harmonic	−70
5th (or more) harmonic	−76

**Table 2 sensors-21-04347-t002:** Comparison of EBPSK modulation schemes.

Scheme	K	N	θ	$f_{c}$	η_max_	Data Rate	Sideband PSD Attenuation
1st	1	10	π	50 kHz	1/34	5000 bps	−20 dB
2nd	1	20	π	50 kHz	1/20	2500 bps	−25 dB
3rd	2	20	π	50 kHz	1/22	2500 bps	−20 dB
4th	2	20	π	40 kHz	1/15	2000 bps	−20 dB

## Data Availability

Not applicable.

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
