# Peer review of "Application of Ultra Narrow Band Modulation in Enhanced Loran System"

_sensors, 2021, doi:10.3390/s21134347_

Round 1

Reviewer 1 Report

The article is about the actual topic, the LORAN system variants are important as the backup system for the navigation application in the case, when the satellite systems are suppresed or crashed (like Gallileo system a few years ago). The motivation of the authors is increasing of the data transmittion rate, (POINT 1:)but in the introduction is missing information, why it is so important, there is missing the analysis if the actual system has any issues with the data transmittion. Sure their work can be used for the realisation of the additional serivces, but description is not include in the article (the possible services and their benefits for the safety of the system).

POINT 2: Line 54: I think that part of the sentence is missing. 

POINT 3: Chapter 2 is not introduced, the chapter 2.1 is immediately after the chapter name. Describe, what is in this chapter.

POINT 4: Maybe, you can support the figure 9, if you will add the block diagram of the system and propagation way and place the markers for the signals from the graphs inside this diagram.

Wish you a good luck with your research.

Reviewer 2 Report

Corrections are given in the PDF document.

Reviewer 3 Report

Dear Editor,

my comments as below:

  1. abstract, please add information about your major findings. What is a novelty of paper? Please add obtained results from research test.
  2. Introduction, please add much more References about application of eLoran system. 
  3. Introduction, last paragraph must included information about novelty of paper. What is a major performance of this paper?
  4. Figure 6, please much better desrcibe the results from Figure 6.
  5. Figure 7, 8, 9 , the same comment like in point 4.
  6. Discussion, please compare the presented research method with respect to References list in Introduction chapter. Why your solution is better?
  7. Where is the chapter of conclusion? I don't see it.
  8. References, see comment in point 2.

Round 2

Reviewer 3 Report

I accept the paper.